# Automatized Detection and Categorization of Fissure Sealants from Intraoral Digital Photographs Using Artificial Intelligence

**DOI:** 10.3390/diagnostics11091608

**Published:** 2021-09-03

**Authors:** Anne Schlickenrieder, Ole Meyer, Jule Schönewolf, Paula Engels, Reinhard Hickel, Volker Gruhn, Marc Hesenius, Jan Kühnisch

**Affiliations:** 1Department of Conservative Dentistry and Periodontology, University Hospital, Ludwig-Maximilians University Munich, 80336 Munich, Germany; anne.schlickenrieder@t-online.de (A.S.); juleschoenewolf@web.de (J.S.); paula.engels@icloud.com (P.E.); hickel@dent.med.uni-muenchen.de (R.H.); 2Institute for Software Engineering, University of Duisburg-Essen, 45147 Essen, Germany; ole.meyer@uni-due.de (O.M.); gruhn@adesso.de (V.G.); marc.hesenius@uni-due.de (M.H.)

**Keywords:** pit and fissure sealants, caries assessment, visual examination, clinical evaluation, artificial intelligence, convolutional neural networks, deep learning, transfer learning

## Abstract

The aim of the present study was to investigate the diagnostic performance of a trained convolutional neural network (CNN) for detecting and categorizing fissure sealants from intraoral photographs using the expert standard as reference. An image set consisting of 2352 digital photographs from permanent posterior teeth (461 unsealed tooth surfaces/1891 sealed surfaces) was divided into a training set (*n* = 1881/364/1517) and a test set (*n* = 471/97/374). All the images were scored according to the following categories: unsealed molar, intact, sufficient and insufficient sealant. Expert diagnoses served as the reference standard for cyclic training and repeated evaluation of the CNN (ResNeXt-101-32x8d), which was trained by using image augmentation and transfer learning. A statistical analysis was performed, including the calculation of contingency tables and areas under the receiver operating characteristic curve (AUC). The results showed that the CNN accurately detected sealants in 98.7% of all the test images, corresponding to an AUC of 0.996. The diagnostic accuracy and AUC were 89.6% and 0.951, respectively, for intact sealant; 83.2% and 0.888, respectively, for sufficient sealant; 92.4 and 0.942, respectively, for insufficient sealant. On the basis of the documented results, it was concluded that good agreement with the reference standard could be achieved for automatized sealant detection by using artificial intelligence methods. Nevertheless, further research is necessary to improve the model performance.

## 1. Introduction

The availability of artificial intelligence (AI) methods has aroused increasing interest in developing convolutional neural networks (CNNs) for automatized detection and categorization of diagnostic images in medicine and dentistry to objectify the classification of pathological findings [1]. In dentistry, radiographs are mostly used as image sources for CNNs to identify pathologies. Specifically, caries detection has been trained on bitewings [2,3,4,5,6,7], apical radiographs [8] or panoramic X-rays [9]. By contrast, there have been few attempts to apply AI technology to assess clinical images, which can be interpreted as a machine-readable equivalent for visual inspection. This study is the first report of automatic detection and categorization of dental caries [10,11,12,13] or dental plaque [14] from clinical photographs. When considering the broad spectrum of pathological findings on dental hard tissue, e.g., caries, erosion or developmental disorders, as well as dental interventions, e.g., sealants, dental restorations or prosthodontic measures, it is evident that CNNs need to be trained separately for each of the aforementioned categories. The aim of this pioneering project on the automatized detection of dental materials was to identify and categorize opaque sealants, which is primarily justified by the frequent use of these sealants in dental health services of industrialized nations [15]. Second, sealant materials constitute a uniform group of materials that are typically white and easily visually detectable on posterior teeth compared to other dental restorations. Consequently, it can be hypothesized that the learning of a CNN for detecting sealants from dental photographs represents a first step before considering other types of dental restorations. Therefore, in this diagnostic study, the ability of a CNN to detect and categorize fissure sealants was investigated (as a test method) using digital photographs of posterior teeth, and the diagnostic outcome was compared with expert evaluation (the reference standard).

## 2. Materials and Methods

### 2.1. Study Design

The reporting of this study followed the recommendations of the Standard for Reporting of Diagnostic Accuracy Studies (STARD) steering committee [16] and topic-related recommendations [17].

### 2.2. Photographic Images

All the images were taken for use in previous studies, as well as for clinical or teaching purposes, by an experienced dentist (J.K.). All the images were photographed using a professional single reflex lens camera (Nikon D300, D7100 or D7200 with a Nikon Micro 105-mm lens; Nikon, Tokyo, Japan) and Macro Flash EM-140 DG (Sigma, Rödermark, Germany) after tooth cleaning and drying. Molar teeth were photographed indirectly using intraoral mirrors (Reflect-Rhod, Hager and Werken, Duisburg, Germany) that were heated before being positioned in the oral cavity to prevent condensation on the mirror surface.

To ensure the best possible image quality, deficient photographs, e.g., out-of-focus images or images with saliva contamination, were excluded. Furthermore, duplicate photos from identical teeth or surfaces were removed from the dataset. This selection step ensured there were no repetitions in the included clinical photographs. All jpeg images (RGB format, resolution 1200 × 1200 pixel, no compression) were cropped to an aspect ratio of 1:1 and/or rotated in a standard manner using professional image editing software (Affinity Photo, Serif, Nottingham, UK) until, finally, the tooth surface filled most of the frame. Considering the study aim, images from healthy teeth or sealed surfaces were also included. Photographs with (additional) cavitated caries lesions or other hard tissue defects, e.g., enamel hypomineralization, hypoplasia, extensive tooth wear, and direct and indirect restorations, were excluded. Finally, 2352 anonymized, high-quality clinical photographs from permanent posterior teeth and the corresponding occlusal surfaces were included.

### 2.3. Categorization of Sealants (Reference Standard)

Each image was examined on a computer to detect and categorize fissure sealants using well-accepted international classification systems [18,19]. The following categories were used: 0—occlusal surfaces with no sealant; 1—occlusal surfaces with a clinically intact fissure sealant (up to one third loss of material in the periphery of the fissure pattern); 2—occlusal surface with a sufficient fissure sealant (retention of the material in the main fissure or loss of material exceeding one third of the fissure pattern); 3—insufficient (nearly complete loss of material and re-exposure of the main fissures) (Figure 1). Each of the given diagnostic categories is typically linked with different treatment modalities in daily dental practice and, in consequence, the quality staging appears of clinical relevance and justifies its scientific consideration in the present study. All the images were prelabeled by a group of three graduated dentists and subsequently independently counterchecked by an experienced examiner (J.K., >20 years of clinical practice and scientific experience). In the case of divergent opinions, each image was discussed until a consensus was reached. Each diagnostic decision—one per image—served as a reference standard for cyclic training and repeated evaluation of the deep-learning-based CNN.

All the annotators (A.S., J.S., P.E.) were trained during a 2-day workshop by the principal investigator (J.K.) and calibrated before beginning the study. The intra- and inter-examiner reproducibility was determined using 60 photographs, and the corresponding Kappa values showed at least a substantial capability for detecting and categorizing fissure sealants. The intra-/inter-examiner reproducibilities were 0.784/0.753 (A.S.), 0.779/0.752 (J.S.) and 0.779/0.752 (P.E.).

### 2.4. Programming and Configuration of the Deep-Learning-Based CNN for Sealant Detection and Categorization (Test Method)

The CNN was trained stepwise using a pipeline of established procedures, mainly image augmentation and transfer learning. Before training, the entire image set (2352 images/461 unsealed tooth surfaces/1891 sealed surfaces) was divided into a training set (*n* = 1881/364/1517) and a test set (*n* = 471/97/374). The latter was never made available to the CNN as training material and served as an independent test set.

Image augmentation was used to provide a large number of variable images to the CNN on a recurring basis. For this purpose, the randomly selected images (batch size = 16) were multiplied by a factor of ~5, altered by image augmentation (random center and margin cropping by up to 30% each, random deletion up to 30%, random affine transformation up to 180 degrees, random perspective transformation up to a distortion of 0.5, and random changes in brightness, contrast and saturation up to 10%) and resized (to 300 × 300 pixels) by using torchvision (version 0.9.1, https://pytorch.org) in conjunction with the PyTorch library (version 1.8.1, https://pytorch.org). All the images were normalized to compensate for under- and overexposure.

ResNeXt-101–32x8d [20] was used as the basis for the continuous adaptation of CNN for sealant detection and categorization. The CNN was trained using backpropagation to determine the gradient for learning. Backpropagation was repeated iteratively for images and labels using the abovementioned batch size and parameters. Overfitting was prevented by first selecting a low learning rate (0.00005) and then performing dropout (at a rate of 0.5) on the final linear layers as a regularization technique [21]. To train the CNN, this step was repeated for 10 epochs. The cross entropy loss as an error function and the Adam optimizer (Betas 0.9 and 0.999, Epsilon 10^−8^) were applied.

To accelerate the training process of the CNN, an open-source neural network with pretrained weights was employed (ResNeXt-101-32x8d pretrained on ImageNet., Stanford Vision and Learning Lab, Stanford University, Palo Alto, CA, USA). This step enabled the transfer of existing learning results to increase the efficiency of recognition of basic structures in the existing image set. The training of the CNN was executed on a university-based server with the following specifications: RTX A6000 48 GB (Nvidia, Santa Clara, CA, USA), i9 10850K 10x3.60 GHz (Intel Corp., Santa Clara, CA, USA) and 64 GB RAM.

### 2.5. Statistical Analysis

The data were analyzed using Python (http://www.python.org, version 3.8). The overall diagnostic accuracy (ACC = (TN + TP)/(TN + TP + FN + FP)) was determined by calculating the number of true positives (TP), false positives (FP), true negatives (TN) and false negatives (FN) after using 25%, 50%, 75% and 100% of the images of the training data set. The sensitivity (SE), specificity (SP), positive and negative predictive values (PPV and NPV, respectively), and the area under the receiver operating characteristic (ROC) curve (AUC) were computed for the selected types of teeth and surfaces [22]. Saliency maps were plotted to identify image areas that are important for the CNN to make individual decisions. We calculated the saliency maps [23] by backpropagating the CNN prediction and visualized the gradient of the input of the resized images (300 × 300 pixels).

## 3. Results

The trained deep-learning-based CNN detected sealants correctly in 98.7% of all the test cases, corresponding to an AUC of 0.996 (Table 1, Figure 2). Additionally, the SE (96.9), SP (99.2), PPV (96.9) and NPV (99.2) were documented to be close to perfect (Table 1). By comparison, the model diagnostic performance was lower for the sealant subcategories (Table 1, Figure 2). Here, the AUC values were highest for the identification of intact sealants (0.951), followed by insufficient sealants (0.942) and sufficient sealants (0.888). These numbers, as well as the other performance data (Table 1), indicate that the automated identification of the subcategories in the present stage was less accurate than the simple detection of opaque sealant material from clinical photographs. The detailed case distribution was obtained from the confusion matrix (Figure 3). Here, the majority of incorrect decisions by the CNN occurred for categories other than the true classification, which indicates there were no major misclassifications. Most incorrect decisions were made for sufficient sealants. This observation is in line with the diagnostic parameters shown in Table 1. In addition to the descriptive and explorative data presentation, saliency maps (Figure 4) were plotted to illustrate the parts of each image that were used by the CNN for decision making.

## 4. Discussion

The results of the present diagnostic study demonstrated that AI algorithms can detect and categorize sealants from machine-readable intraoral photographs. A high diagnostic accuracy of 98.7% and AUC of 0.996 were found (Table 1). Unlike this promising result, the CNN classified subcategories less accurately. Here, a diagnostic performance of approximately 90% accuracy was achieved (Table 1). In particular, sufficiently sealed occlusal surfaces were identified less reliably than the two other categories, which illustrates that further improvement is needed.

In addition, it can be concluded that the developed CNN can be used in future software applications and can identify sealants accurately with a high probability from intraoral photographs. To our knowledge, no comparable studies have been carried out thus far on the evaluation of fissure sealants using artificial intelligence, which should be recognized as a unique feature of our study. The current diagnostic performance data fit into the overall context of existing dental studies. For example, studies with a similar methodology have documented an accuracy of up to 90% for the detection of caries lesions from clinical images [10,11] or radiographs [2,3,4,5,6,7,8,9]. Considering earlier published data from methodologically similar projects, it can be concluded that our most recent results (Table 1, Figure 2 and Figure 3) are in line with an expected outcome. Our data need to be critically assessed from different methodological perspectives. First, it should be highlighted that the pipeline used for image augmentation, transfer learning and the chosen CNN architecture (ResNeXt-101-32x8d) represents an up-to-date approach that may have enhanced the documented results. Second, as our study was performed on good quality professional clinical photographs, the results may have been be positively influenced by this factor. None of the images used were overexposed or underexposed, and the teeth investigated were mostly free of plaque, calculus and saliva. All the images were normalized, cropped and standardized before processing. Third, only unsealed posterior teeth and sealed teeth of varying quality were included in the study materials. Cases with caries lesions, developmental defects, and direct or indirect dental restorations were excluded from the project to enable unbiased learning of the CNN. Another methodological advantage in this context appears to be the use of single tooth images, because interfering information from adjacent teeth or margins was mostly excluded. Consequently, it can be expected that the use of other image formats, e.g., clinical images with multiple teeth or the whole jaw, will result in a lower model performance. The number of available clinical photographs is a limitation that must be critically examined. Here, several thousand images at best should be includable, as the number of images is a crucial consideration for this type of study. In the present analysis, we were able to include 2352 clinical images, which should be interpreted as the minimum number. This fact should not be underestimated, because increasing the number of images will extend the training of the CNN and could improve the CNN precision. Further improvements in the model performance can be expected by extending the number of image samples and using the image segmentation technique. The latter approach results in precise image labeling and could be considered as the method of choice to reach the long-term goal of almost perfect detection and assessment of fissure sealants from clinical photographs by AI methods.

## 5. Conclusions

The clinical application of AI methods in software applications may be feasible but fundamental dental research needs to be performed first. The results of the present study show that a trained CNN detected sealant intraoral photographs with an agreement of 98.7% with reference decisions. The categorical classification into intact, sufficient and insufficient sealants was performed with a diagnostic accuracy of approximately 90%. Considering the complexity of intraoral findings, it can be concluded that further training of AI-based detection, as well as categorization of prevalent and less-prevalent dental diseases and all types of restorations, is required before clinical use can be recommended.

## Figures and Tables

**Figure 1 diagnostics-11-01608-f001:**
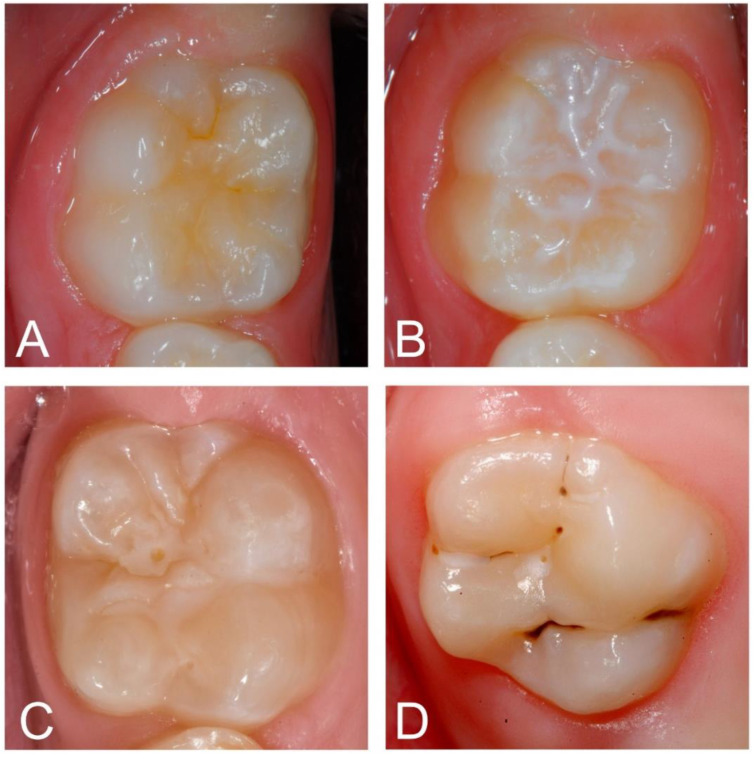
Example clinical images for each category: unsealed molar (**A**) and intact (**B**), sufficient (**C**) and insufficient fissure sealant (**D**).

**Figure 2 diagnostics-11-01608-f002:**
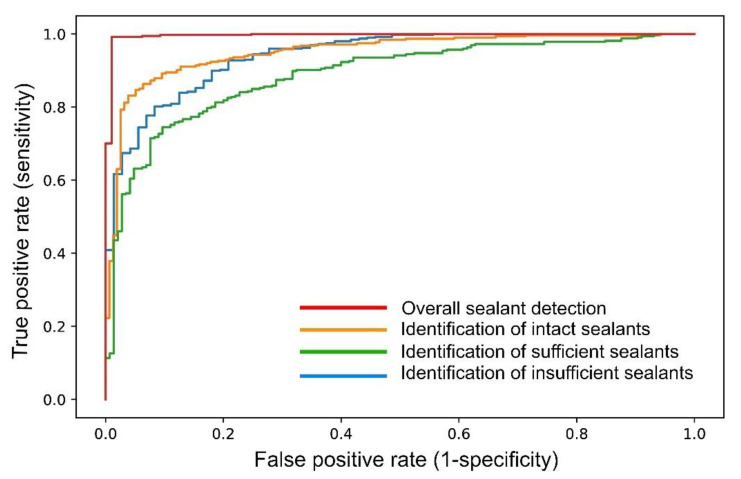
The ROC curves illustrate the model performance of the developed CNN for overall and categorical sealant detection.

**Figure 3 diagnostics-11-01608-f003:**
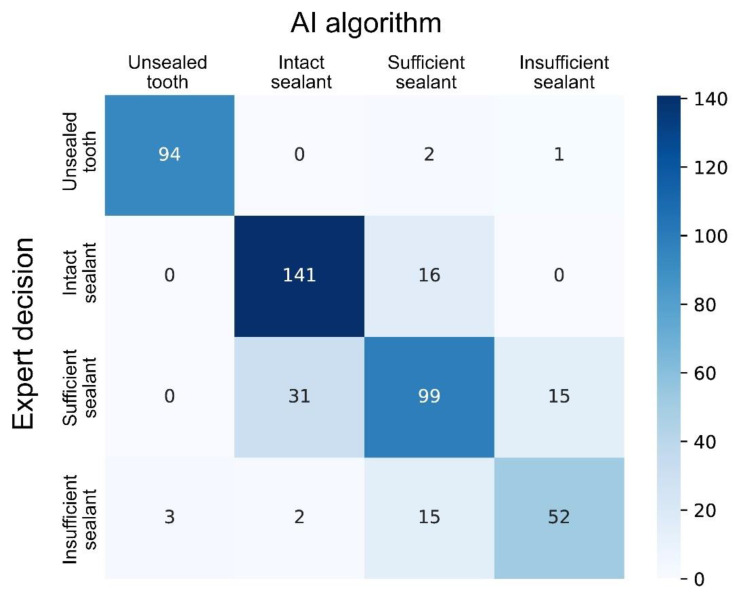
Confusion matrix showing the CNN classification performance for the test sample.

**Figure 4 diagnostics-11-01608-f004:**
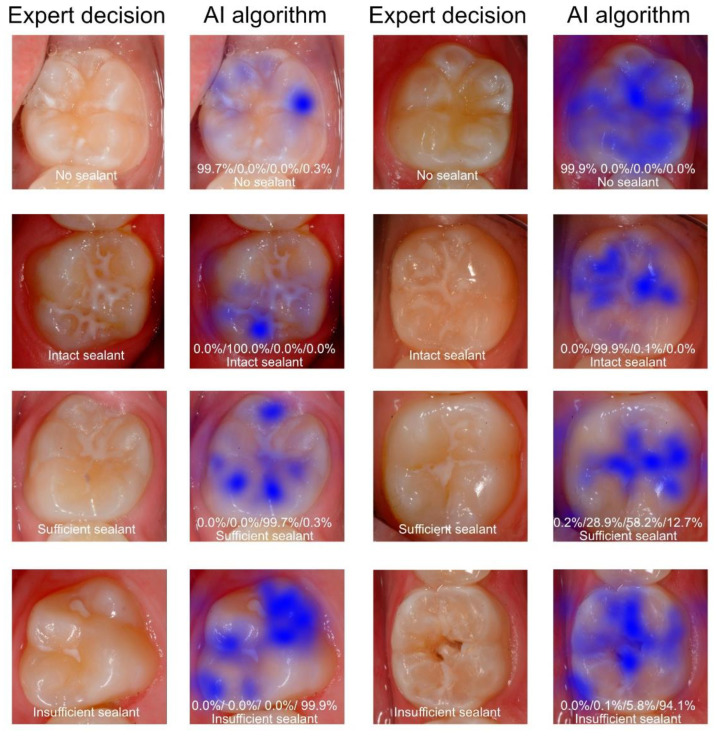
Examples of clinical images showing the reference decision and paired images with saliency maps visualizing the image areas (in blue) used in decision making by the AI method. The corresponding test results by the AI method are given for each example (unsealed tooth/intact fissure sealant/sufficient sealant/insufficient sealant).

**Table 1 diagnostics-11-01608-t001:** Overview of the diagnostic performance of the developed convolutional neural network (CNN), where the independent test set (*n* = 471) was compared against independent expert evaluation of the caries detection level. The calculations were performed for different types of teeth, surfaces and training steps. In this context, the overall diagnostic accuracy (ACC), sensitivity (SE), specificity (SP), negative predictive value (NPV), positive predictive value (PPV) and area under the receiver operating characteristic curve (AUC).

Diagnostic Categories	True Positives (TP)		True Negatives (TN)		False Positives (FP)		False Negatives (FN)		Diagnostic Performance					
	*n*	%	*n*	%	*n*	%	*n*	%	ACC	SE	SP	NPV	PPV	AUC
Overall sealant detection	94	20.0	371	78.8	3	0.6	3	0.6	98.7	96.9	99.2	99.2	96.9	0.996
Identification of intact sealants	141	29.9	281	59.7	33	7.0	16	3.4	89.6	89.8	89.5	94.6	81.0	0.951
Identification of sufficient sealants	99	21.0	293	62.2	33	7.0	46	9.8	83.2	68.3	89.9	86.4	75.0	0.888
Identification of insufficient sealants	52	11.0	383	81.3	16	3.4	20	4.3	92.4	72.2	96.0	95.0	76.5	0.942

## Data Availability

The datasets used and analyzed during the current study are available from the corresponding author on reasonable request.

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
