# Peer review of "Automatized Detection and Categorization of Fissure Sealants from Intraoral Digital Photographs Using Artificial Intelligence"

_diagnostics, 2021, doi:10.3390/diagnostics11091608_

Round 1

Reviewer 1 Report

Using a CNN network, good agreement with the reference standard could be achieved for automatized sealant detection by using artificial intelligence methods. Sample size was large enough. Method was sound. Presentation was clear.

One issue is that the processing time was not reported. It should be provided so that readers can better understand how this can be incorporated into clinical practice.

Author Response

Thank you for your positive evaluation! 

Reviewer 2 Report

This is a well-written manuscript and the authors present an interesting work.

It is meaningful that it is the first attempt to diagnose the presence or absence of sealant at the time when AI is introduced into the diagnostic field.

It seems that the completion of the manuscript can be further improved if a few corrections are added. 

  1. The author used an intraoral digital photograph to conduct this study, but the title of the paper does not match the content as an "intraoral radiograph".
  2. Table1 -  The full name for abbreviation is missing.
  3. Why did the authors conduct the trial in four categories? Is the classification clinically meaningful? It will be helpful to understand if more information on this is added.

Author Response

This is a well-written manuscript and the authors present an interesting work.

Response:          Thank you for reviewing and the positive evaluation! ?

It is meaningful that it is the first attempt to diagnose the presence or absence of sealant at the time when AI is introduced into the diagnostic field.

It seems that the completion of the manuscript can be further improved if a few corrections are added. 

  1. The author used an intraoral digital photograph to conduct this study, but the title of the paper does not match the content as an "intraoral radiograph".

Response:      Thank you for detecting the error. We modified the title

Revised text:  Title.

  1. Table1 -  The full name for abbreviation is missing.

Response:      Thank you for this point. We modified the legend and included the full name for each abbreviation.

Revised text:  Table 1/ legend.

  1. Why did the authors conduct the trial in four categories? Is the classification clinically meaningful? It will be helpful to understand if more information on this is added.

Response:      Thank you for this remark. A wide range in retention of pit and fissure sealants can be observed in clinical dental practice and each category is linked with different treatment modalities. In consequence, the quality staging is of clinical relevance and justifies its consideration in the present study.

Revised text:  Line 88-91
